# Beneficial Effect of Sirolimus-Pretreated Mesenchymal Stem Cell Implantation on Diabetic Retinopathy in Rats

**DOI:** 10.3390/biomedicines12020383

**Published:** 2024-02-07

**Authors:** Nanyoung Kang, Ji Seung Jung, Jiyi Hwang, Sang-Eun Park, Myeongjee Kwon, Haerin Yoon, Jungyeon Yong, Heung-Myong Woo, Kyung-Mee Park

**Affiliations:** 1Laboratory of Veterinary Ophthalmology, College of Veterinary Medicine, Chungbuk National University, Cheongju 28644, Republic of Korea; rkdsksdud1@gmail.com (N.K.); wjdwltmd00@gmail.com (J.S.J.); wldml1013@gmail.com (J.H.); sangeunpark1992@gmail.com (S.-E.P.); k3clover@naver.com (M.K.); yun3038@naver.com (H.Y.); gold5181@naver.com (J.Y.); 2Laboratory of Veterinary Surgery, College of Veterinary Medicine, Kangwon National University, Chuncheon 24341, Republic of Korea; woohm@kangwon.ac.kr

**Keywords:** diabetic retinopathy, sirolimus, mesenchymal stem cells, subconjunctival injection

## Abstract

Background: Diabetic retinopathy (DR) is a vision-threatening complication that affects virtually all diabetic patients. Various treatments have been attempted, but they have many side effects and limitations. Alternatively, stem cell therapy is being actively researched, but it faces challenges due to a low cell survival rate. In this study, stem cells were pretreated with sirolimus, which is known to promote cell differentiation and enhance the survival rate. Additionally, the subconjunctival route was employed to reduce complications following intravitreal injections. Methods: Diabetes mellitus was induced by intraperitoneal injection of 55 mg/kg of streptozotocin (STZ), and DR was confirmed at 10 weeks after DM induction through electroretinogram (ERG). The rats were divided into four groups: intact control group (INT), diabetic retinopathy group (DR), DR group with subconjunctival MSC injection (DR-MSC), and DR group with subconjunctival sirolimus-pretreated MSC injection (DR-MSC-S). The effects of transplantation were evaluated using ERG and histological examinations. Results: The ERG results showed that the DR-MSC-S group did not significantly differ from the INT in b-wave amplitude and exhibited significantly higher values than the DR-MSC and DR groups (*p* < 0.01). The flicker amplitude results showed that the DR-MSC and DR-MSC-S groups had significantly higher values than the DR group (*p* < 0.01). Histological examination revealed that the retinal layers were thinner in the DR-induced groups compared to the INT group, with the DR-MSC-S group showing the thickest retinal layers among them. Conclusions: Subconjunctival injection of sirolimus-pretreated MSCs can enhance retinal function and mitigate histological changes in the STZ-induced DR rat model.

## 1. Introduction

Diabetes mellitus (DM) is the most common endocrine disease in humans. As reported by the International Diabetes Federation in 2021, approximately 537 million adults (ages 20–79) have diabetes, constituting about 10.5% of the total global adult population. The prevalence of DM is continuously rising, with projections exceeding 780 million cases by 2045 [1]. DM leads to microvascular abnormalities, resulting in extensive damage to systemic tissues, including the eyes. Diabetic retinopathy (DR), a chronic microvascular complication of DM, impairs vision and affects nearly all patients with DM. Approximately 2% of all individuals with DM experience blindness due to DR, making it the leading cause of vision loss in adults over the age of 25 [2,3].

There are various methods to create a DM model, among which the administration of streptozotocin (STZ) is a commonly used approach. STZ induces the destruction of pancreatic β cells and is widely used experimentally to create a type 1 diabetes model [4,5]. STZ-induced DR typically develops after prolonged exposure to hyperglycemia levels above 150 mg/dL [6].

Exposure to DM conditions leads to various biochemical and metabolic abnormalities, including changes in the redox state of pyridine nucleotides, the accumulation of sorbitol, over-activation of protein kinase C, oxidative stress due to excessive free radical production, and alterations in hemodynamics. These pathogenic mechanisms play a crucial role in the progression of DR. Additionally, within weeks of the onset of diabetes, leukostasis occurs in the retinal capillaries, leading to capillary occlusion and local ischemia [7]. Retinal hypoxia results in the increased expression of vascular endothelial growth factor (VEGF) [8]. Elevated levels of VEGF induce angiogenesis and enhance retinal vessel permeability, causing disruption of the barrier between the retina and blood [8,9].

Current treatments for DR include retinal photocoagulation using lasers and intravitreal injections of anti-VEGF agents. However, due to its destructive nature, laser photocoagulation can result in permanent damage to retinal cells [10,11]. Anti-VEGF therapy has shown superior results in reducing vision loss and improving the rate of vision recovery compared to laser monotherapy. However, its effects are short-lived, necessitating continuous follow-up observation and injection therapy [10,12,13,14]. Additionally, cases of non-responsiveness to this therapy also occasionally occur. These methods target only vascular pathology, hence highlighting the need for the development of new treatments with different mechanisms [15,16].

Mesenchymal stem cells (MSCs) have the ability to differentiate into various cell lineages, thereby promoting tissue regeneration and enhancing function [17]. Furthermore, through paracrine effects, they can secrete immunomodulatory, anti-angiogenic, and neurotrophic factors [18]. They also support mitochondrial function, which is crucial in restoring retinal cell functionality [19]. Additionally, MSCs inhibit the secretion of pro-inflammatory cytokines and reduce oxidative damage [20]. Numerous studies have shown MSCs to be effective in treating retinal diseases, demonstrating their ability to prevent retinal capillary dropout, loss of ganglion cells, oxidative damage, and neovascularization [16,21,22].

However, the application of stem cells alone faces a significant challenge due to their low survival and adhesion rates [19]. This issue is particularly pronounced in hyperglycemic conditions, where an excessive accumulation of reactive oxygen species (ROS) can alter the regenerative abilities of MSCs, leading to decreased survival rates and reduced efficacy [23].

Therefore, there are attempts to enhance the success rate of stem cell therapies through pre-conditioning/treatment of the cells [24]. One such approach involves subjecting the stem cells to be transplanted to conditions similar to the harsh microenvironment of damaged tissues, such as hypoxia, thereby improving the cells’ resistance to the stress of the host environment [25,26]. Another method involves pretreatment with drugs; there are already studies that have enhanced the efficacy of stem cells by pretreating them with tacrolimus, dexamethasone, and sirolimus [19,27,28,29].

Sirolimus, also known as rapamycin, was initially isolated from the bacterium *Streptomyces hygroscopicus* [30]. This drug functions by inhibiting the mammalian target of rapamycin, a key regulator of cell growth, proliferation, survival, protein synthesis, and autophagy [31,32]. Studies have shown that sirolimus enhances autophagy, regulates energy metabolism, reduces oxygen consumption, and ROS production and thereby promotes stem cell differentiation, increasing cell migration and survival rates [27,28].

Thus, the aim of this study is to investigate the therapeutic effects of sirolimus-pretreated MSC transplantation in the treatment of the STZ-induced DR rat model.

## 2. Materials and Methods

### 2.1. Animals

The research conducted was approved by the Institutional Animal Care and Use Committee (IACUC) at Chungbuk National University (Approval No. CBNUA-2032-22-01), in accordance with the Association for Research in Vision and Ophthalmology statement for the Use of Animals in Ophthalmic and Vision Research.

Twenty-five male Sprague-Dawley rats, aged 8 weeks, were obtained from Nara Biotech (Pyeongtaek, Republic of Korea). They were housed in a conventional environment with a standard 12 h light/12 h dark cycle. The rats had free access to normal pellet chow (Experimental Rat & Mouse Diet, Purina, St. Louis, MO, USA) and water.

### 2.2. Measurements of Body Weight (BW) and Blood Glucose (BG)

All BW and BG measurements were conducted after a 6-h fasting period. The initial dataset was collected just before the diabetes induction. The second and third datasets were measured at 3 weeks and 10 weeks post-diabetes induction, respectively. The final measurements were taken at week 17 post-diabetes induction.

### 2.3. Diabetes Induction

DM was induced in the rats after an 8-h fasting period. This was achieved through an intraperitoneal injection of STZ (Sigma-Aldrich, St. Louis, MO, USA) in citrate buffer (pH 4.5) (Sigma-Aldrich) at a dose of 55 mg/kg. The intact control group (INT, eight rats, sixteen eyes) received an equivalent volume of citrate buffer via intraperitoneal injection [33].

DM was confirmed twenty-three days post-STZ injection when the BG exceeded 250 mg/dL, as measured by a commercial blood glucose meter (FORA G11, ForaCare, Moorpark, CA, USA). The seventeen diabetic rats were randomly divided into three groups: the diabetic retinopathy group (DR, seven rats, fourteen eyes); the DR group with subconjunctival MSC injection (DR-MSC, four rats, eight eyes); and the DR group with subconjunctival sirolimus-pretreated MSC injection (DR-MSC-S, six rats, twelve eyes).

### 2.4. Preparations and Injections of MSCs

Human umbilical cord blood-derived mesenchymal stem cells (hUCB-MSCs) were obtained from Kang Stem Biotech (Seoul, Republic of Korea). These cells were cultured using a KSB-3 Complete Medium^®^ Kit (Kang Stem Biotech) with 10% fetal bovine serum (Thermo Fisher Scientific Inc., Waltham, MA, USA) in a humidified atmosphere containing 5% CO_2_ at 37 °C. For the DR-MSC-S group, 100 nM of sirolimus dissolved in dimethyl sulfoxide (DMSO) (Sigma-Aldrich) was pretreated in the culture media for 24 h [19,34]. The same amount of DMSO, without sirolimus, was added to the culture media for the same duration for the DR-MSC group.

After 10 weeks of DM induction, subconjunctival MSC injections (1 × 10^5^ MSCs in 10 µL phosphate-buffered saline; PBS) and sirolimus-pretreated MSC injections were performed using a 31G insulin syringe (Ultra-Fine II short needle, BD Biosciences, Franklin Lakes, NJ, USA). General anesthesia was induced by isoflurane (Terrell, Piramal Critical Care, Bethlehem, PA, USA) and topical anesthesia was induced by proparacaine (Alcaine, Alcon, Geneva, Switzerland). Then, disinfection of the surface of the globe was performed using 0.5% povidone iodine. In the INT and DR groups, 10 µL of PBS was injected using the same protocol. These injections were repeated twice, with an eight-day interval between administrations.

### 2.5. Electroretinography (ERG)

Ten weeks post-STZ injection, all the rats underwent 6 h of dark adaptation. ERG evaluations were then conducted after pupil dilation using topical 0.01% tropicamide and phenylephrine (Mydrin-P, Santen, Osaka, Japan). Flash and flicker stimuli (8.0 cd·s/m^2^ at 2 Hz and 28.3 Hz, respectively) were utilized with the RETevet ERG system (LKC, Gaithersburg, MD, USA). The examination aimed to identify the presence of DR. Subsequent ERG assessments were performed fourteen weeks post-STZ injection to evaluate retinal function using the same protocol.

### 2.6. Histological Evaluation

Eighteen weeks post-STZ injection, three rats from each group were sacrificed, and their eyes were immediately removed and immersed in BioFix HD (BioGnost, Zagreb, Croatia). The eyes were bisected along the optic nerve, creating two equal halves, and the lenses and vitreous were removed. Following routine tissue processing, the eyes were then embedded in paraffin. The paraffin-embedded sections were stained using routine hematoxylin and eosin (H&E) staining. The thickness of the retinal tissue was measured at a magnification ×200.

### 2.7. Statistical Analysis

The data were analyzed using Prism 10 software (GraphPad Software, Boston, MA, USA). The results are presented as the mean ± standard deviation. Statistical significance between the groups was determined using an ordinary one-way analysis of variance. *p* values less than 0.01 were considered statistically significant.

## 3. Results

### 3.1. Assessment of BW and BG

The final measurements were taken at week 17 post-diabetes induction. A comparison of the BW and BG data was conducted across the INT, DR, DR-MSC, and DR-MSC-S groups. The averages of the BW for the INT, DR, DR-MSC, and DR-MSC-S groups were 589.50 ± 36.75 g, 234.20 ± 36.68 g, 262.50 ± 22.05 g, and 231.00 ± 26.76 g, respectively (Figure 1A). The averages of the BG were 98.75 ± 11.62 mg/dL, 540.70 ± 134.70 mg/dL, 429.50 ± 48.00 mg/dL, and 534.50 ± 102.70 mg/dL, respectively (Figure 1B). Prior to STZ administration, there were no significant differences in the BW and BG among the groups. However, post-diabetes induction by STZ, the INT group showed a significantly higher BW and a lower BG compared to the diabetic groups. No significant differences in the BW and BG were observed among the DR, DR-MSC, and DR-MSC-S groups at week 17. This suggests that subconjunctival administration of MSCs and sirolimus-pretreated MSCs did not demonstrate systemic therapeutic effects in DM.

### 3.2. Confirmation of DR with ERG

To confirm the induction of DR prior to the subconjunctival injections of the substance, ERG assessments were conducted at 10 weeks post-diabetes induction. The average flash b-wave amplitude in the INT group and the diabetes-induced groups was 153.2 ± 56.59 µV and 73.84 ± 21.25 µV, respectively (Figure 2A). The average flicker amplitude in the INT group and the diabetes-induced groups was 118.59 ± 32.29 µV and 66.41 ± 24.28 µV, respectively (Figure 2B). Both results were significantly lower in the diabetes-induced groups compared to the INT group (*p* < 0.01), confirming the induction of DR.

### 3.3. Retinal Function Evaluation with ERG

ERG measurements were repeated at 14 weeks post-diabetes induction, two weeks after the final substance administration, to evaluate changes in retinal function. The average flicker amplitudes in the INT, DR, DR-MSC, and DR-MSC-S groups were 147.00 ± 28.62 µV, 48.28 ± 17.03 µV, 88.27 ± 23.91 µV, and 107.30 ± 23.01 µV, respectively (Figure 3A). The DR-MSC group showed significantly higher values compared to the DR group (*p* < 0.01). The DR-MSC-S group demonstrated significantly higher values than both the DR and DR-MSC groups (*p* < 0.01), with no statistical difference from the INT group.

The average flash b-wave amplitudes in the INT, DR, DR-MSC, and DR-MSC-S groups were 173.30 ± 46.19 µV, 60.21 ± 22.31 µV, 89.53 ± 18.46 µV, and 148.50 ± 17.32 µV, respectively (Figure 3B). The DR-MSC-S group exhibited significantly higher values than both the DR and DR-MSC groups (*p* < 0.01).

### 3.4. Histological Evaluation of the Retina

Histological evaluation was conducted 7 weeks after the initial subconjunctival injections, which was 18 weeks post-diabetes induction. All the retinas were examined using H&E staining (Figure 4A). The total retinal thicknesses for the INT, DR, DR-MSC, and DR-MSC-S groups were 237.20 ± 13.71 µm, 179.60 ± 16.18 µm, 191.80 ± 18.78 µm, and 215.90 ± 12.04 µm, respectively. A decrease in the total retinal thickness was observed in the diabetic groups compared to the INT group. The DR-MSC-S group exhibited a significantly greater retinal thickness compared to the DR and DR-MSC groups (*p* < 0.01).

The thickness of the inner nuclear layer (INL) for the INT, DR, DR-MSC, and DR-MSC-S groups was 38.70 ± 5.93 µm, 31.21 ± 5.30 µm, 37.17 ± 7.64 µm, and 38.45 ± 6.84 µm, respectively (Figure 4C). The thickness of the photoreceptor layer (PRL) for the INT, DR, DR-MSC, and DR-MSC-S groups was 33.13 ± 4.19 µm, 25.43 ± 3.56 µm, 29.5 ± 5.13 µm, and 32.17 ± 4.93 µm, respectively (Figure 4D). Both layer thicknesses were significantly greater in the INT and DR-MSC-S groups compared to the DR group (*p* < 0.01).

## 4. Discussion

This study has demonstrated that subconjunctival injection of sirolimus-pretreated MSCs can increase the b-wave amplitude and flicker amplitude in ERG recordings from rats with DR while histologically mitigating the thinning of the retinal layers. These findings suggest that subconjunctival injections of sirolimus-pretreated MSCs may be therapeutically effective for DR.

Previous research indicates that DR develops three months after the intraperitoneal injection of STZ in rats, accompanied by thinning of the retinal layers and an increase in neovascularization [35,36]. In this study, a significant reduction in the ERG amplitude compared to the INT group was observed 10 weeks post-STZ injection, indicating the occurrence of DR, which was further supported by histological assessments showing reduced thickness of the total retinal layer.

Patients with DR are known to experience significant decreases in retinal function compared to their pre-disease state [37]. According to various studies measuring ERG in diabetic rats, the DR-induced groups exhibited a significantly reduced b-wave compared to the normal control [38,39,40,41]. Furthermore, flicker ERG recorded at a frequency of 30 Hz is reduced in amplitude in patients with moderate to severe DR [42]. Consistent with these findings, this study demonstrated a substantial decrease in both b-wave and flicker ERG amplitude in the DR-induced groups compared to the INT group. Nevertheless, an improvement in both amplitudes was observed in the DR-MSC-S group compared to the DR group, suggesting that sirolimus-pretreated MSCs may improve the retinal function of DR rats.

Histological changes in DR include thinning of the retinal layers, loss of retinal cells, formation of neovascularization, and increased inflammation [43,44]. Additionally, reductions in the thickness of the ganglion cell layer, INL, and PRL have been observed [45,46,47,48]. These changes suggest neurodegeneration and an increase in inflammatory and neurodegenerative markers has been reported in rats with DM induced by STZ [43]. Consistent with previous research, this study also observed a decrease in the total retinal thickness in the diabetic groups. However, the overall retinal thickness was greater in the DR-MSC-S group than in the DR and DR-MSC groups, and the thicknesses of the INL and PRL were also greater in the DR-MSC-S group compared to the DR group. These results indicate that subconjunctival injection of sirolimus-pretreated MSCs can mitigate histopathological changes in the retina.

A challenge in treating diabetic patients with MSCs is the reduced survival rate of MSCs in a hyperglycemic environment. Research has shown that MSCs cultured in serum from type 2 diabetic patients exhibit significantly decreased cell survival [49]. Various studies indicate that hyperglycemic conditions can lead to an increase in mitochondrial glucose metabolism, which through mitochondrial hyperpolarization induces the production of ROS [50,51,52]. Additionally, a chronic hyperglycemic environment can lead to the upregulation and activation of cyclooxygenase, which may increase not only oxidative stress but also inflammatory responses [53,54]. Excessive accumulation of ROS leads to mitochondrial damage, cell apoptosis, inflammation, and lipid peroxidation [55]. Consequently, persistently high glucose concentrations alter the potential regenerative capacity of MSCs and ultimately reduce their survival rate, lowering the success rate of treatment [19]. In this study, the DR-MSC group did not show significant differences from the DR group in the flash b-wave amplitude or histological evaluations, which could be attributed to the low survival rate of MSCs in a hyperglycemic environment.

In this study, the pretreating of stem cells with sirolimus, a technique already proven effective in several studies, was used. In mice, sirolimus has been shown to enhance autologous regeneration and hematopoiesis in hematopoietic stem cells [32]. Treatment with sirolimus-pretreated MSCs in a systemic lupus erythematosus mouse model alleviated clinical symptoms and extended survival, also enhancing the immunomodulatory function of MSCs [26]. Furthermore, pretreatment with sirolimus significantly increased autophagy activity and lysosome production in cells and reduced cell apoptosis under harsh conditions compared to untreated cells. The post-transplantation of sirolimus-pretreated cells markedly improved the repair and functional recovery in infarcted myocardium [56]. Additionally, numerous studies have used sirolimus via direct intraocular injection for retinal treatment, confirming its intraocular stability [57,58,59,60]. However, the concentration of sirolimus used in this study was significantly lower than that shown to be effective in previous research, suggesting a lower likelihood of direct effects from sirolimus.

Sirolimus can modulate inflammatory responses related to the generation of ROS and nitric oxide in cells [61]. Sirolimus-pretreated MSCs have shown increased survival or growth factor secretion in hypoxic and serum-deprivation conditions, increased production of survival or growth factors post-transplantation, and suppressed production of inflammatory cytokines [28]. These actions may account for the significantly higher results obtained by the DR-MSC-S group in the ERG and histological evaluations compared to the DR and DR-MSC groups.

Moreover, this study adopted the subconjunctival route instead of the commonly used intravitreal injection for DR treatment. Intravitreal injection has the advantage of delivering drugs directly to the vitreous and retina, but the technique is invasive with risks of increased intraocular pressure, infection, inflammation, potential damage to the lens, retinal toxicity, and detachment [62,63]. However, subconjunctival injection is less invasive and carries a lower risk of complications associated with intravitreal injections, making repeated administration feasible. Subconjunctival injections are generally thought to primarily deliver drugs to the anterior segment of the eye, but there are studies showing that effective drug delivery to the vitreous and retinal/vitreous layers is also possible [64,65]. Administering drugs or placing implants in the subconjunctival space can skip the barriers of the conjunctiva and cornea, resulting in enhanced permeability in the retina/choroidal region [63]. Nevertheless, subconjunctival injection results in a lower drug concentration reaching the posterior segment compared to intravitreal injection. There is a study showing that administering MSCs at weekly intervals is more effective in inhibiting disease progression than a single administration [66]. This study adopted a method of repeated administrations at 8-day intervals for a total of two times to enhance the therapeutic efficacy. Consequently, this study found subconjunctival injection to be effective in treating DR, with minimal systemic impact on the BW or BG.

In this study, the effectiveness of sirolimus-pretreated MSCs was evaluated, focusing on the functional improvement of the retina as measured by ERG and the histological assessment of retinal layer thickness. However, the evaluation was confined to a maximum duration of 7 weeks, indicating the necessity for further research into the long-term effects. Another limitation of this study is the absence of a comparative analysis between different concentrations of pretreated sirolimus as well as a lack of comparison between single and repeated administrations.

## 5. Conclusions

In conclusion, this study has demonstrated that DR rats treated with sirolimus-pretreated MSCs exhibit an increase in ERG amplitude. Additionally, there was a mitigation in the reduction of the retinal thickness at a histological level. These results suggest that sirolimus-pretreated MSCs can enhance retinal function and mitigate histological changes in rats with DR, indicating their potential for application in DR treatment. They also suggest that the subconjunctival route of administration can be effective for the treatment of retinopathy.

## Figures and Tables

**Figure 1 biomedicines-12-00383-f001:**
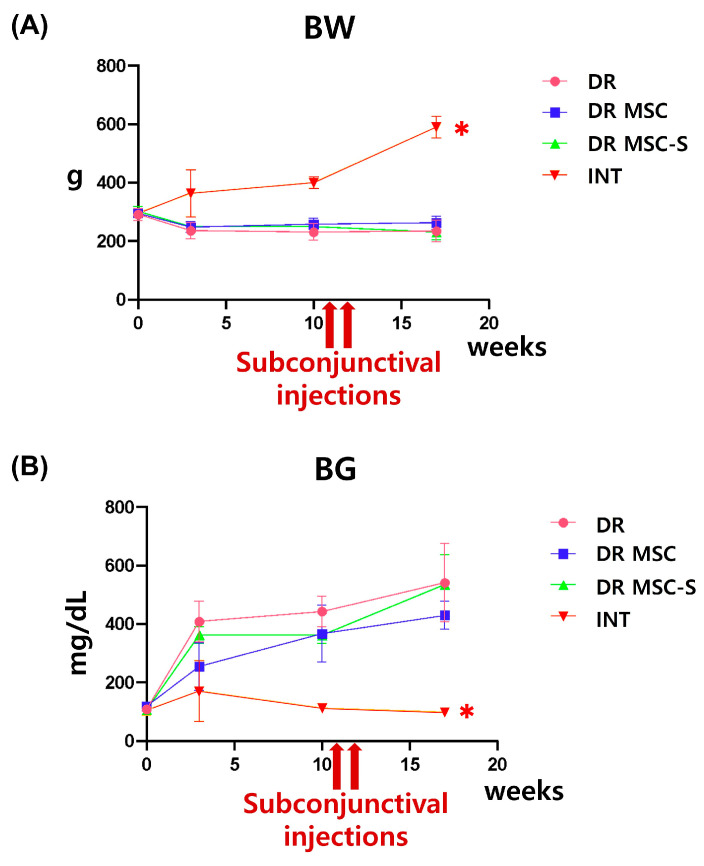
Changes in BW and BG over time. (**A**) A graph for BW changes over time (**B**) A graph for BG changes over time; subconjunctival administration was conducted at 11 and 12 weeks post-diabetes induction. There were no significant differences in BW and BG among the DR, DR-MSC, and DR-MSC-S groups at week 17. The INT group exhibited significantly higher BW and lower BG at all time points. * *p* < 0.01, INT, n = 8 rats; DR, n = 7 rats; DR-MSC, n = 4 rats; DR-MSC-S, n = 6 rats. Abbreviations: BW, body weight; BG, blood glucose; INT, intact control group; DR, diabetic retinopathy group; DR-MSC, DR group with subconjunctival MSC injection; DR-MSC-S, DR group with subconjunctival sirolimus-pretreated MSC injection.

**Figure 2 biomedicines-12-00383-f002:**
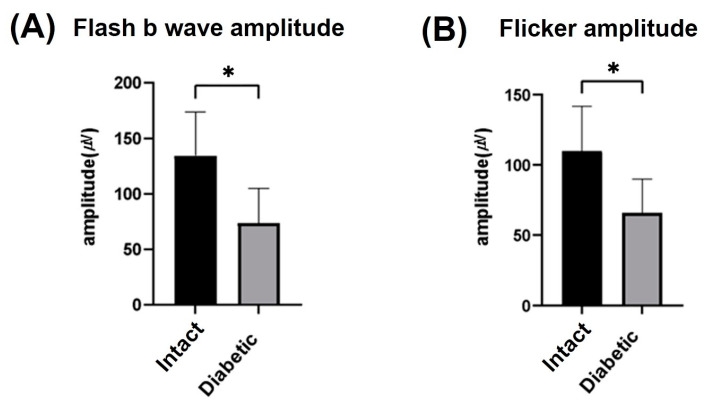
Comparison of electroretinogram results between intact and diabetic groups at week 10 post-diabetes induction. (**A**) Results of flash b-wave amplitude and (**B**) Results of flicker amplitude at 10 weeks post-diabetes induction; the diabetic group showed significantly lower values compared to the intact group, indicating the induction of diabetic retinopathy. * *p* < 0.01, Intact, n = 8 rats, 16 eyes; Diabetic, n = 17 rats, 34 eyes.

**Figure 3 biomedicines-12-00383-f003:**
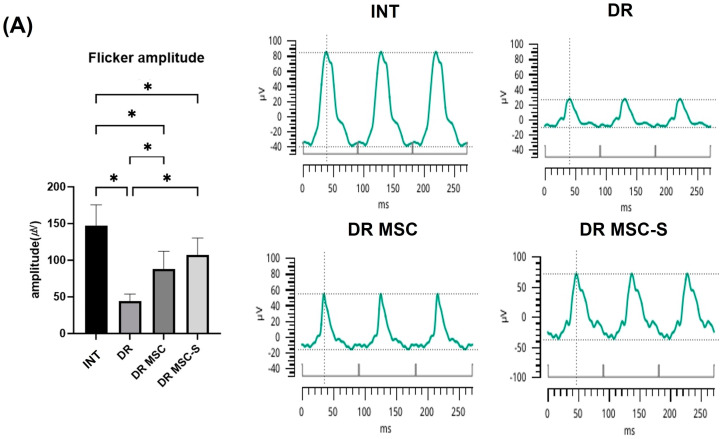
Comparative analysis of flicker and flash b-wave amplitudes in ERG across groups. (**A**) Flicker stimulus ERG results at 14 weeks post-diabetes induction; the DR-MSC and DR-MSC-S groups exhibited significantly greater amplitude than the DR group. (**B**) Flash stimulus ERG results at 14 weeks post-diabetes induction; the DR-MSC-S group showed significantly greater amplitude than both the DR and DR-MSC groups and did not differ statistically from the INT group. * *p* < 0.01. INT, n = 8 rats, 16 eyes; DR, n = 7 rats, 14 eyes; DR-MSC, n = 4 rats, 8 eyes; DR-MSC-S, n = 6 rats, 12 eyes. Abbreviations: ERG, electroretinogram; INT, intact control group; DR, diabetic retinopathy group; DR-MSC, DR group with subconjunctival MSC injection; DR-MSC-S, DR group with subconjunctival sirolimus-pretreated MSC injection.

**Figure 4 biomedicines-12-00383-f004:**
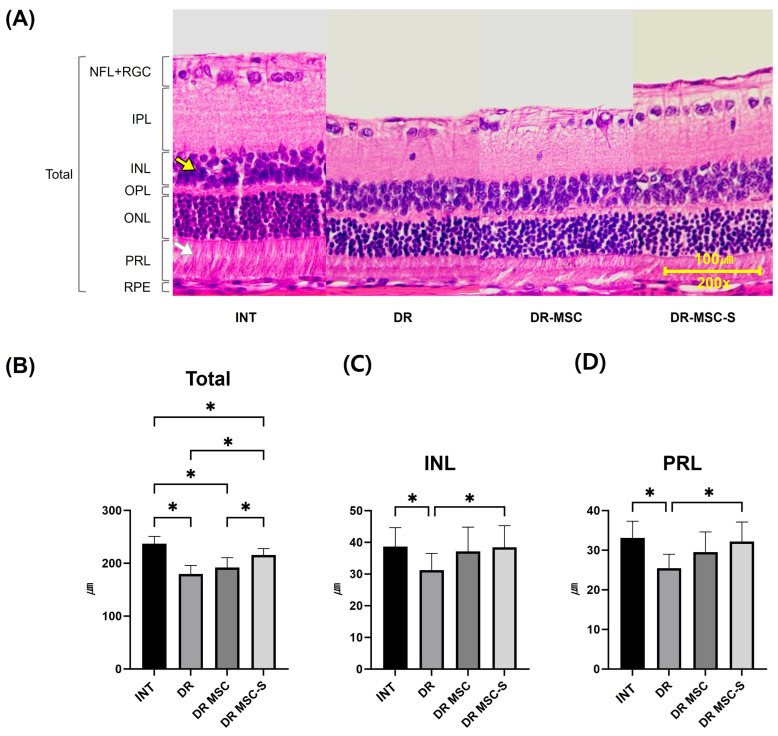
Comparison of histological evaluations of retina using H&E staining. (**A**) Total retinal layer of each group 7 weeks after the initial subconjunctival injections, which was 18 weeks post-diabetes induction (**B**) Results of total retinal thickness measuring; the DR-MSC-S had a significantly thicker retina compared to DR and DR-MSC groups. (**C**) Results of INL (yellow arrow) thickness measuring. The DR had the thinnest inner nuclear layer that showed significant difference with INT and DR-MSC-S. There were no statistical differences between INT, DR-MSC, and DR-MSC-S. (**D**) Results of PRL (white arrow) thickness measuring. The DR had the thinnest PRL that showed significant difference with INT and DR-MSC-S. There were no statistical differences between INT, DR-MSC, and DR-MSC-S. H&E staining, magnification ×200. * *p* < 0.01. INT, n = 8 rats, 16 eyes; DR, n = 7 rats, 14 eyes; DR-MSC, n = 4 rats, 8 eyes; DR-MSC-S, n = 6 rats, 12 eyes. Abbreviations: H&E, hematoxylin and eosin; NFL, nerve fiber layer; RGC, retinal ganglion cell layer; IPL, inner plexiform layer; INL, inner nuclear layer; OPL, outer plexiform layer; ONL, outer nuclear layer; PRL, photoreceptor layer; INT, intact control group; DR, diabetic retinopathy group; DR-MSC, DR group with subconjunctival MSC injection; DR-MSC-S, DR group with subconjunctival sirolimus-pretreated MSC injection.

## Data Availability

The datasets analyzed during the current study are available from the corresponding author upon reasonable request.

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
