# Peer review of "Beneficial Effect of Sirolimus-Pretreated Mesenchymal Stem Cell Implantation on Diabetic Retinopathy in Rats"

_biomedicines, 2024, doi:10.3390/biomedicines12020383_

Round 1

Reviewer 1 Report

Comments and Suggestions for Authors

The presented paper is devoted to description of the efffect of sirolimus-pretreated mesenchymal stem cell subconjuctival injection on diabetic retinopathy in rats. The topic is very actual and well corresponds to current rersearch in the earea of ocular drug delivery.

The paper is well organized and well prepared. It could be accepted after minor revisions.

Comments:

1. Authors should explain why they didn't use OCT for DR control.

2. Please add arrows or areas on Figure 4(A) to show the locations of interest.

Author Response

We would like to express our sincere gratitude for your invaluable feedback and constructive comments during the revision process of our manuscript. Your insights and suggestions have significantly contributed to the improvement and overall quality of the paper. Please find the detailed responses below and the corresponding revisions/corrections highlighted/in track changes in the re-submitted files.

Comments 1: Authors should explain why they didn't use OCT for DR control.

Response 1: Thank you for pointing this out. We had considered using Optical coherence tomography (OCT); however, in this study, substance administration was carried out after induction of diabetic retinopathy (DR) in rats. Streptozotocin-induced DR typically occurs after prolonged exposure to high blood glucose levels, generally exceeding 150 mg/dL [1]. In this study, DR was confirmed to be induced at the 10-week of diabetes induction. However, according to Kyselova (2005), cataract progression becomes evident in streptozotocin-induced diabetic rats at the 10-week of diabetes induction [2]. Consistent with this research, cataracts were observed in the majority of diabetic rats in our study. The presence of cataracts impedes accurate imaging during OCT as the light dose not effectively penetrate the eye. Due to the same reasons, evaluating retinal angiography was also challenging. Consequently, in this study, OCT was not utilized for retinal assessment; instead, retinal tissues were collected and evaluated histopathologically.

1. Quiroz, J.; Yazdanyar, A. Animal Models of Diabetic Retinopathy. Ann. Transl. Med. 2017, 93, doi:10.21037/atm-20-6737.2. Kyselová, Z.; Garcia, S.; Gajdošíková, A.; Gajdošík, A.; Štefek, M. Temporal Relationship between Lens Protein Oxidation and Cataract Development in Streptozotocin-Induced Diabetic Rats. Physiol. Res. 2005, 54, 49–56, doi:10.33549/physiolres.930613.

Comments 2: Please add arrows or areas on Figure 4(A) to show the locations of interest.

Response 2: Thank you for enhancing the clarity of my figure. We have made modifications based on your comments to emphasize the relevant locations. The Inner Nuclear Layer (INL) has been indicated with a yellow arrow, while the Photoreceptor Layer (PRL) is represented by a white arrow. Additionally, we have upgraded the figure to a higher definition.

Reviewer 2 Report

Comments and Suggestions for Authors

Dear Editor,

I would like to express my deep thanks to you for allowing me to review this valuable manuscript “Beneficial Effect of Sirolimus-pretreated Mesenchymal Stem Cell Implantation on Diabetic Retinopathy in Rats". The objective of this research is to examine the therapeutic benefits of sirolimus-pre-treated MSC transplantation in the treatment of a rat model with STZ-induced DR..

.

·         The authors need to expand on the pathophysiology of Diabetic Retinopathy in the introduction.

·         The authors created a beautiful figure, but the figure in the manuscript is not clear enough. The original or high-definition figures should be provided

·         Scientific articles with several technical terms or acronyms should include an abbreviation list. Abbreviations can improve article reading and comprehension

·         Figures must be self-explanatory (Fig. 3 and Fig. 4), so abbreviations must be explained as footnotes for readers to view without the need to refer to the main text

Author Response

We would like to express our sincere gratitude for your invaluable feedback and constructive comments during the revision process of our manuscript. Your insights and suggestions have significantly contributed to the improvement and overall quality of the paper. Please find the detailed responses below and the corresponding revisions/corrections highlighted/in track changes in the re-submitted files.

 Comments 1: The authors need to expand on the pathophysiology of Diabetic Retinopathy in the introduction.

Response 1: We appreciate the reviewer’s comment on the lack of pathophysiology of diabetic retinopathy in the introduction. In accordance with the reviewer’s comment, we have revised the manuscript. In the original version, only the microvascular damage caused by hyperglycemia in diabetes, resulting in retinal hypoxia and increased vascular endothelial growth factor (VEGF) levels, was mentioned. In the revised manuscript, various biochemical and metabolic abnormalities associated with diabetic conditions are detailed, leading to retinal hypoxia. As a consequence, an increase in VEGF levels is described, which leads to angiogenesis, increased retinal vessel permeability, and disruption of the barrier between the retina and blood. This modification can be found in the second paragraph on page 2, lines 47-55 of the revised manuscript.

Here's the updated text in the manuscript.

[Exposure to DM conditions leads to various biochemical and metabolic abnormalities, including changes in the redox state of pyridine nucleotides, accumulation of sorbitol, over-activation of protein kinase C, oxidative stress due to excessive free radical production, and alterations in hemodynamics. These pathogenic mechanisms play a crucial role in the progression of DR. Additionally, within weeks of the onset of diabetes, leukostasis occurs in retinal capillary, leading to capillary occlusion and local ischemia [1]. Retinal hypoxia results in increased expression of vascular endothelial growth factor (VEGF) [2]. Elevated levels of VEGF induce angiogenesis and enhance retinal vessel permeability, causing disruption of the barrier between the retina and blood [2,3].

  1. Curtis, T.M.; Gardiner, T.A.; Stitt, A.W. Microvascular Lesions of Diabetic Retinopathy: Clues towards Understanding Pathogenesis? Eye 2009, 23, 1496–1508, doi:10.1038/eye.2009.108.
  2. Pande, G.S.; Tidake, P. Laser Treatment Modalities for Diabetic Retinopathy. Cureus 2022, 14, e30024, doi:10.7759/cureus.30024.
  3. Kollias, A.N.; Ulbig, M.W. Diabetic Retinopathy. Dtsch. Ärzteblatt Int. 2010, 107, 75–83, doi:10.3238/arztebl.2010.0075.

Comments 2:  The authors created a beautiful figure, but the figure in the manuscript is not clear enough. The original or high-definition figures should be provided.

Response 2: We apologize for the poor quality of the figures. We have attached the high-definition figures below and sent the original files via e-mail to [email protected]. Also, I have changed figures in the manuscript to high-definition and add arrows for clarity. This modification can be found in the page 8.

Comments 3:  Scientific articles with several technical terms or acronyms should include an abbreviation list. Abbreviations can improve article reading and comprehension

Response 3: We appreciate the suggestion to include an abbreviation list. We believe that this addition improves the readability and comprehension of our manuscript. Therefore, we have added the abbreviation list in the supplementary material. This can be found in the page 12 of revised manuscript. Additionally, we have attached the list of abbreviations below.

List of abbreviations

BG

Blood glucose

BW

Body weight

DM

Diabetes mellitus

DMSO

Dimethyl sulfoxide

DR

Diabetic retinopathy

DR-MSC

Diabetic retinopathy group with subconjunctival mesenchymal stem cell injection

DR-MSC-S

Diabetic retinopathy group with subconjunctival sirolimus-pretreated mesenchymal stem cell injection

ERG

Electroretinogram

H&E

Hematoxylin and eosin

hUCB-MSCs

Human umbilical cord blood-derived mesenchymal stem cells

INL

Inner nuclear layer

INT

Intact control group

IPL

Inner plexiform layer

MSCs

Mesenchymal stem cells

NFL

Nerve fiber layer

ONL

Outer nuclear layer

OPL

Outer plexiform layer

PBS

Phosphate-buffered saline

PRL

Photoreceptor layer

RGC

Retinal ganglion cell layer

ROS

Reactive oxygen species

STZ

Streptozotocin

VEGF

Vascular endothelial growth factor

Comments 4:  Figures must be self-explanatory (Fig. 3 and Fig. 4), so abbreviations must be explained as footnotes for readers to view without the need to refer to the main text.

Response 4: Thank you for pointing this out. we have added explanation of abbreviations in the foot notes of Figure 1., Figure 3. and Figure 4. These changes can be found in page 5, 7 and 8. 

Reviewer 3 Report

Comments and Suggestions for Authors

The topic is very relevant, the authors proving that subconjunctival injection of sirolimus-pre-treated mesenchymal stem cells can enhance retinal function and mitigate histological changes in induced Diabetic retinopathy in rat 28 model.

The methodology is very modern and complex, using molecular biology, electroretinography and histology, and adequate statistic analysis.

The results have revealed that, subconjunctival injection of sirolimus-pretreated MSCs can increase b-wave amplitude and flicker amplitude in ERG recordings from rats 251 with DR, while histologically mitigating the thinning of retinal layers. These findings sug- 252 gest that subconjunctival injections of sirolimus-pretreated mesenchymal stem cells may be therapeutically effective for Diabetic Retinopathy. These results may open new research directions in treatment of Diabetic Retinopathy.

The conclusions are consistent with the evidence and arguments presented.

The references are very relevant, including also some relevant author’s previous experience in the field.

I suggest some minor editing corrections

1.       References 4 and 32- authors should not be written with capital letters

Author Response

We would like to express our sincere gratitude for your invaluable feedback and constructive comments during the revision process of our manuscript. Your insights and suggestions have significantly contributed to the improvement and overall quality of the paper. Please find the detailed responses below and the corresponding revisions/corrections highlighted/in track changes in the re-submitted files.

 Comments 1:  References 4 and 32- authors should not be written with capital letters

Response 1: We appreciate your feedback and have made the corrections to the authors’ names. These changes are reflected on pages 12 and 13. Additionally, following adjustments based on another reviewer’s comments, reference 32 has been revised to reference 30. We have attached the updated contents below.

4. Rakieten, N.; Rakieten, M.L.; Nadkarni, M.V. Studies on the Diabetogenic Action of Strepto-zotocin (NSC-37917). Cancer Chemother. Rep. 1963, 29, 91–98.

30. Vézina, C.; Kudelski, A.; Sehgal, S.N. RAPAMYCIN (AY-22, 989), A NEW ANTIFUNGAL AN-TIBIOTIC. The Journal of Antibiotics 1975, 28, 721–726, doi:10.7164/antibiotics.28.721.
